

# *Apis andreniformis* associated Actinomycetes show antimicrobial activity against black rot pathogen (*Xanthomonas campestris* pv. *campestris*)

Yaowanoot Promnuan[1], Saran Promsai[1], Wasu Pathom-aree[2,3] and Sujinan Meelai[4]

[1] Department of Microbiology, Faculty of Liberal Arts and Science, Kasetsart University –Kamphaeng Saen campus, Kamphaeng Saen, Nakhon Pathom, Thailand

[2] Department of Biology, Faculty of Science, Chiang Mai University, Chiang Mai, Thailand

[3] Research Center in Bioresources for Agriculture, Industry and Medicine, Faculty of Science, Chiang Mai University, Chiang Mai, Thailand

[4] Department of Microbiology, Faculty of Science, Silpakorn University –Sanam Chandra Palace campus, Nakhon Pathom, Nakhon Pathom, Thailand

## ABSTRACT

This study aimed to investigate cultivable actinomycetes associated with rare honey bee species in Thailand and their antagonistic activity against plant pathogenic bacteria. Actinomycetes were selectively isolated from the black dwarf honey bee (*Apis andreniformis*). A total of 64 actinomycete isolates were obtained with *Streptomyces* as the predominant genus (84.4%) followed by *Micromonospora* (7.8%), *Nonomuraea* (4.7%) and *Actinomadura* (3.1%). All isolates were screened for antimicrobial activity against *Xanthomonas campestris* pv. *campestris, Pectobacterium carotovorum* and *Pseudomonas syringae* pv. *sesame*. Three isolates inhibited the growth of *X. campestris* pv. *campestris* during *in vitro* screening. The crude extracts of two isolates (ASC3-2 and ASC5-7P) had a minimum inhibitory concentration (MIC) of 128 mg L$^{-1}$ against *X. campestris* pv. *campestris*. For isolate ACZ2-27, its crude extract showed stronger inhibitory effect with a lower MIC value of 64 mg L$^{-1}$ against *X. campestris* pv. *campestris*. These three active isolates were identified as members of the genus *Streptomyces* based on their 16S rRNA gene sequences. Phylogenetic analysis based on the maximum likelihood algorithm showed that isolate ACZ2-27, ASC3-2 and ASC5-7P were closely related to *Streptomyces misionensis* NBRC 13063$^T$ (99.71%), *Streptomyces cacaoi* subsp. *cacaoi* NBRC 12748$^T$ (100%) and *Streptomyces puniceus* NBRC 12811$^T$ (100%), respectively. In addition, representative isolates from non-*Streptomyces* groups were identified by 16S rRNA gene sequence analysis. High similarities were found with members of the genera *Actinomadura, Micromonospora* and *Nonomuraea*. Our study provides evidence of actinomycetes associated with the black dwarf honey bee including members of rare genera. Antimicrobial potential of these insect associated *Streptomyces* was also demonstrated especially the antibacterial activity against phytopathogenic bacteria.

Corresponding author
Wasu Pathom-aree,
wasu.p@cmu.ac.th,
wasu215793@gmail.com

## INTRODUCTION

*Apis andreniformis* or black dwarf honey bee is one of the rare native bees found in Thailand and both tropical and near subtropical regions of Southeast Asia (*Hepburn & Radloff, 2011*). They build a single, exposed comb on the small branches of shrubs, banana, bamboos or small trees (*Wongsiri et al., 1996*). Actinomycetes are Gram-positive bacteria which produce two types of branching mycelia, namely aerial and substrate mycelium. They are well known as prolific producer of bioactive compounds, especially members of the *Streptomyces* species. Actinomycetes are widely distributed in nature both in terrestrial and aquatic environments (*Goodfellow & Williams, 1983*). They are recently found to be associated with several insects such as ants (*Currie et al., 1999*; *Currie et al., 2003*; *Cafaro & Currie, 2005*; *Oh et al., 2009*; *Van Arnam et al., 2016*; *Chevrette et al., 2019*), leafcutter bees (*Inglis, Sinler & Goette, 1993*), wasp (*Kroiss et al., 2010*; *Matarrita-Carranza et al., 2021*), southern pine beetle (*Scott et al., 2008*), honey bees and stingless bees (*Promnuan, Kudo & Chantawannakul, 2009*; *Promnuan, Kudo & Chantawannakul, 2011*; *Promnuan, Promsai & Meelai, 2020*).

There are few studies on actinomycetes associated with honey bee (*Apis*) species in Thailand. Actinomycetes belonging to the genera *Streptomyces*, *Nonomurea* and *Nocardiopsis* were isolated from three species of honey bees (*Apis mellifera*, *Apis cereana* and *Apis florea*) collected from northern Thailand. Furthermore, some of these isolates were able to inhibit the growth of *Paenibacillus larvae* and *Melisococcus plutonius in vitro* (*Promnuan, Kudo & Chantawannakul, 2009*). A novel actinobacterial species, *Actinomadura apis* was isolated from an *A. mellifera* hive in Chiang Mai province, Thailand (*Promnuan, Kudo & Chantawannakul, 2011*). In addition, *Streptomyces* spp. isolated from giant honey bee (*Apis dorsata*) showed the ability to inhibit the growth of *Xanthomonas oryzae* pv. *oryzae*, *Xanthomonas campestris* pv. *campestris*, *Ralstonia solanacearum* and *Pectobacterium carotovorum* (*Promnuan, Promsai & Meelai, 2020*). These studies indicated that honey bees harbored actinomycetes with potential to produce novel antimicrobial compounds to combat bacterial diseases in agriculture. Nevertheless, currently the discovery rate of new antibiotics from actinomycetes has been declining especially those from common habitats. Therefore, the focus of search and discovery programs for novel antibiotics from actinomycetes has shifted toward unexplored habitats (*Bundale et al., 2019*; *Rangseekaew & Pathom-aree, 2019*). In addition, rare actinomycetes are of interest for drug discovery programs as they are producers of many antibiotics in the market including rifamycins (*Amycolatopsis mediterranei*) and erythromycin (*Saccharopolyspora erythraea*). During the last two decades, known antibiotics produced by rare actinomycetes were increased up to 25–30% (*Ding et al., 2019*). The Gram-negative bacterium, *X. campestris* pv. *campestris*, is known to cause significant losses in many crop plants from diseases such as tomato speck, rice and pomegranate bacterial blight, citrus canker and brassica black rot (*Yan et al., 2019*). This bacterium can cause disease in a large number of species in the Brassicaceae, including *Brassica* and *Arabidopsis*. Black rot is a seed-borne disease and typical symptoms include V-shaped yellow lesions starting from the leaf margins and blackening of the veins (*Vicente & Holub, 2013*). The use of bactericidal compounds to control phytopathogenic

bacteria such as antibiotics and copper could cause serious problems to human health and environment such as antibiotic resistance and toxicity. Furthermore, some emerging strains have shown strong resistance to the antibiotics (*Satish, Raveesha & Jandrdhana, 2002*; *Sabir et al., 2017*; *Mougou & Boughalleb-M'hamdi, 2018*; *Wu et al., 2019*). For these reasons, other control measures have been developed and reported including antagonistic bacteria (*Bacillus* spp. and *Pseudomonas* spp.) (*Wulff et al., 2002*; *Mishra & Arora, 2012*), antimicrobial compounds from plant extracts (*Satish, Raveesha & Jandrdhana, 2002*; *Kaur et al., 2016*) and essential oils (*Sabir et al., 2017*; *Amini et al., 2018*).

Actinomycetes isolated from the black dwarf honey bee (*A. andreniformis*) with antagonistic activity against the phytobacterial pathogen have never been studied and reported. Therefore, this study focused on the isolation of cultivable actinomycetes from *A. andreniformis* collected from Chiang Mai province, Thailand and their antimicrobial activity against phytopathogenic bacteria was investigated.

## MATERIALS & METHODS

### Sample collection
This study was approved by the Institutional Animal Care and the Use Committee (IACUC), Silpakorn University (Ethic number: 8603.16/0328). *A. andreniformis* combs were obtained from the local villages in Mae-rim district, Chiang Mai province, Thailand during October 2013–January 2014. The permission was received from the farm owners (Mr. Ton Tatiya and Mr. Ma Madamun). Six combs were stored in sterilized containers and transferred back to the laboratory. The samples (adult bees, pupae, honey and pollen) were kept in sterile plastic tubes and stored at −20 °C until further processing.

### Isolation and characterization of actinomycetes from *A. andreniformis*
Actinomycetes were isolated from six combs of *A. andreniformis*. Five adult bees and pupae were taken from each of the six collected combs, surface sterilized and ground before the isolation process. Isolation of actinomycetes from one milliliter of honey and one gram of pollen was achieved using a standard 10-fold dilution spread plate method on glycerol-asparagine (ISP5) (*Pridham & Lyons, 1961*), starch casein nitrate agar (*Küster & Williams, 1964*), Czapek's agar (*Waksman, 1950*) and nutrient agar supplemented with 25 μg mL$^{-1}$ of nalidixic acid and nystatin. All plates were incubated at 30 °C for 14–21 days. Presumptive actinomycetes colonies were purified and maintained on yeast extract-malt extract agar (ISP2) (*Shirling & Gottlieb, 1966*) slants and kept in 4 °C. Sixty-four actinomycetes were grouped based on their morphological characteristics in particular the colour of substrate mycelium, aerial spore mass and diffusible pigments.

### Phytopathogenic bacteria
The bacterial phytopathogens, *X. campestris* pv. *campestris* and *P. carotovorum* were obtained from the Department of Agriculture, Ministry of Agriculture and Cooperative, Thailand. *Pseudomonas syringae* pv. *sesame* TISTR 901 was obtained from Thailand Institute of Scientific and Technological Research (TISTR), Thailand. These phytopathogenic bacteria were activated and maintained on nutrient agar and yeast extract-malt extract (ISP2) agar for 24–48 h, at 30 °C before use.

## Screening for antagonistic activity of actinomycetes against plant pathogenic bacteria

All actinomycete isolates were tested for their activity against three plant pathogenic bacteria (*X. campestris* pv. *campestris*, *P. carotovorum* and *Ps. syringae* pv. *sesame* TISTR 901) using a modified cross streak method as described by *Promnuan, Promsai & Meelai (2020)* on glucose yeast extract (GYE) and ISP2 agar plate. The inhibition zones were recorded after 24 h. Each experiment was performed in triplicate.

## Extraction of bioactive compounds from potent actinomycete isolates

Selected actinomycete isolates that showed potent activity against the growth of *X. campestris* pv. *campestris* from the cross-streaking method were grown on ISP2 agar plates (for ACZ2-27) and GYE agar plates (for ASC3-2 and ASC5-7P) and incubated at 30 °C. After 14 days incubation, the agar media of each isolate was extracted using ethyl acetate followed the method as described by *Promnuan, Promsai & Meelai (2020)*. Briefly, small pieces of agar medium (approximately 0.5 cm × 0.5 cm) were added to 200 ml of ethyl acetate and shaken at 150 rpm, 30 °C for 48 h. The extracts were filtered through Whatman No. 1 filter paper and concentrated in a rotary evaporator at 40 °C. One microliter of sterile dimethyl sulfoxide (DMSO) was added into the dried extracts and stored at −20 °C.

## Determination of minimum inhibitory concentration (MIC) of antagonistic actinomycete isolates

The MIC of ethyl acetate extract was determined using modified broth microdilution method in 96-well microtiter plate as described by *Wiegand, Hilpert & Hancock (2008)*. The concentration of the test organism, *X. campestris* pv. *campestris* was adjusted to 0.5 McFarland standard. The initial concentration of the crude extract was adjusted to 1,280 mg $L^{-1}$ using sterile DMSO. The crude extract was prepared as two-fold dilution series using ISP2 broth in a 96-well microtiter plate. The concentration of the crude extract was ranged from 0.25 to 128 mg $L^{-1}$. Sterile DMSO was used as a negative control. After incubation at 30 °C for 24 h, the suspension from each well was inoculated onto an ISP2 agar plate using streak plate technique. Plates were incubated at 30 °C for 24 h. The lowest concentration of the extracts that showed no bacterial growth on the ISP2 agar plate was recorded as the MIC value. This experiment was conducted in triplicate.

## Identification of actinomycetes using 16S rRNA gene

The representative isolates of each morphological group and the actinomycetes that showed strong activity against *X. campestris* pv. *campestris* were grown in ISP2 broth, incubated at 30 °C for 7 days on the rotary shaker (120 rev $min^{-1}$). The cells were collected by centrifugation (91,000 g) and washed three times using sterile distilled water. Genomic DNA extraction and PCR amplification of the 16S rRNA gene were carried out as described by *Nakajima et al. (1999)* using primers 20F (5′-AGTTTGATCCTGGCTC) and 1540R (5′-AAGGAGGTGATCCAGCC). The PCR product was purified using Invitrogen™ PureLink™ PCR Purification Kit (Thermo Fisher Scientific, USA). The 16S rRNA genes were sequenced by 1st BASE, Singapore using the Sanger method. BLAST analysis of actinomycete isolates was done using the EzBioCloud database

(https://www.ezbiocloud.net/) (Yoon et al., 2017). Multiple alignment of the closely related type strain sequences obtained from the GenBank database were carried out using Clustal_W in BioEdit Sequence Alignment Editor 7.2.5 (Hall, 1999). The maximum likelihood (ML) trees were constructed using MEGA X version 10.1.8 (Kumar et al., 2018) based on a comparison of 1,299-1,386 nucleotides present in all the strains used after elimination of gaps and ambiguous nucleotides from the sequences. *Streptomyces thermocarboxydus* DSM 44293[T] was used as an outgroup. Bootstrap analyses based on 1,000 resamplings was used to determine the confidence values for branches of the maximum likelihood (ML) tree (Felsenstein, 1985). The percentage of sequence similarity were calculated using the pairwise alignments function in BioEdit 7.2.5.

## RESULTS

### Isolation and characterization of the actinomycetes from *A. andreniformis*

Adult bees, pupae, honey and pollen were obtained from six combs of the black dwarf honey bee (*A. andreniformis*). A total of sixty-four actinomycetes were isolated from different media as summarized in Table 1. Most actinomycetes were obtained from adult bees (65.6%) followed by honey (20.3%), pollen (12.5%) and pupae (1.6%). Based on the morphological characteristics, all actinomycete isolates were assigned to four groups (Table 2). *Streptomyces* (group IV) was the predominant genus with 84.4% followed by *Micromonospora* (group I) (7.8%), *Nonomuraea* (group III) (4.7%) and *Actinomadura* (group II) (3.1%). *Streptomyces* (ACZ2-27 and ASC3-2) and all rare genera were recovered from the adult honey bee (Table 2).

### Antimicrobial activity of actinomycetes against phytobacterial pathogens

Three isolates from group IV (ACZ2-27, ASC3-2 and ASC5-7P) showed strong inhibition of the growth of a phytobacterial pathogen. Actinomycete isolate ACZ2-27 inhibited the growth of *X. campestris* pv. *campestris* on an ISP2 agar plate (Fig. 1A) with an inhibition zone diameter of 10.75 ± 0.83 mm. The actinomycete isolates ASC3-2 and ASC5-7P inhibited the growth of *X. campestris* pv. *campestris* on a GYE agar plate (Figs. 1B, 1C) with an inhibition zone diameter of 6.00 ± 0.71 and 9.0 ± 0.82 mm, respectively. The MIC levels of ethyl acetate extracts of the three isolates were determined against the growth of *X. campestris* pv. *campestris* using the 96-well microtiter assay. The MIC value of the crude extract of ACZ2-27 against *X. campestris* pv. *campestris* was 64 mg L$^{-1}$ and the MIC values of the crude extracts of ASC3-2 and ASC5-7P were 128 mg L$^{-1}$. These results provide the first evidence of antibacterial extracts from black dwarf honey bee associated actinomycetes against the black rot pathogen, *X. campestris* pv. *campestris*.

### Identification of actinomycetes using 16S rRNA gene

Three actinomycete isolates which showed strong inhibition against the growth of *X. campestris* pv. *campestris* were identified based on the 16S rRNA gene sequence analysis. The 16S rRNA gene sequences of strains ACZ2-27 (LC500236), ASC3-2 (LC506284) and

**Table 1  Number of actinomycetes isolated from *A. andreniformis* using four different media.**

| Sample source | ISP5* | SC* | CZ* | NA* | Total (%) |
|---|---|---|---|---|---|
| | Isolation medium | | | | |
| Adults | 17 | 14 | 6 | 5 | 42 (65.6%) |
| Pupae | 1 | 0 | 0 | 0 | 1 (1.6%) |
| Pollen | 2 | 5 | 1 | 0 | 8 (12.5%) |
| Honey | 9 | 4 | 0 | 0 | 13 (20.3%) |
| Total (%) | 29 (45.3%) | 23 (35.9%) | 7 (10.9%) | 5 (7.8%) | 64 (100%) |

Notes.
*ISP5, Glycerol asparagine agar; SC, Starch casein nitrate agar; CZ, Czapek agar and NA, Nutrient agar.

**Table 2  Assignment of actinomycete isolates into four groups (I-IV) and 16S rDNA identification of the representative isolates.**

| Group | Morphological characteristics | Sources | Occurrence (%) | Representative isolates | Accession no. | Genus |
|---|---|---|---|---|---|---|
| I | wrinkled colony with brown or orange colour | Adult | 7.8 | AGA3-9 | LC546088 | *Micromonospora* |
| | | Adult | | AGA3-53 | LC546089 | |
| II | convex and rigid colony with cream colour | Adult | 3.1 | AGA3-58 | LC546090 | *Actinomadura* |
| III | wrinkled colony with cream colour | Adult | 4.7 | ASC2-5 | LC546091 | *Nonomuraea* |
| IV | powdery colonies and aerial spore mass with white or grey | Adult | 84.4 | ACZ2-27 | LC500236 | *Streptomyces* |
| | | Adult | | ASC3-2 | LC506284 | |
| | | Pollen | | ASC5-7P | LC506285 | |

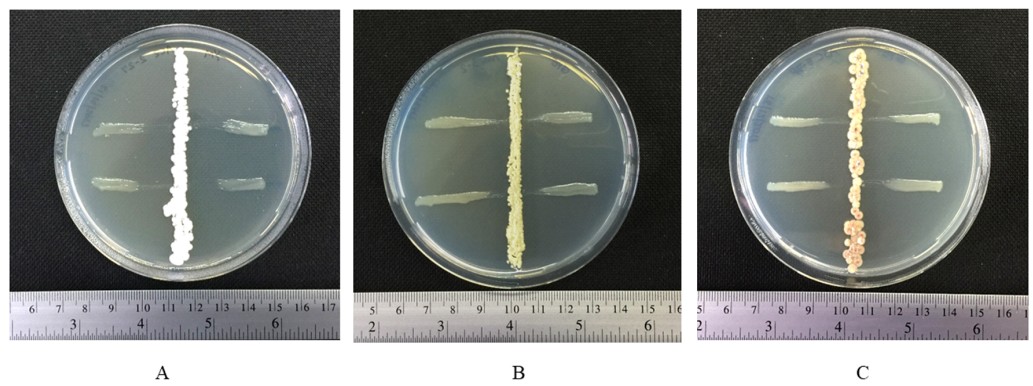

**Figure 1  Inhibitory effect of actinomycetes against growth of *X. campestris* pv. *campestris*.** Fig. 1 Inhibitory effect of actinomycetes against growth of *X. campestris* pv. *campestris*: (A) ACZ2-27 on ISP2 agar plate and (B) ASC3-2 and (C) ASC5-7P on GYE agar plates.

ASC5-7P (LC506285) were analyzed by BLAST in the EzBioCloud database. All strains exhibited high similarities with members of the genus *Streptomyces*. The 16S rRNA gene sequences of all three isolates were compared with the corresponding sequences of the most closely related strains of the genus *Streptomyces*. The maximum likelihood tree (Fig. 2) revealed that strain ACZ2-27, ASC3-2 and ASC5-7P were closely related to *S. misionensis* NBRC 13063[T], *S. cacaoi* subsp. *cacaoi* NBRC 12748[T] and *S. puniceus* NBRC 12811[T]. The 16S rRNA gene sequence similarity percentage between each isolate and their

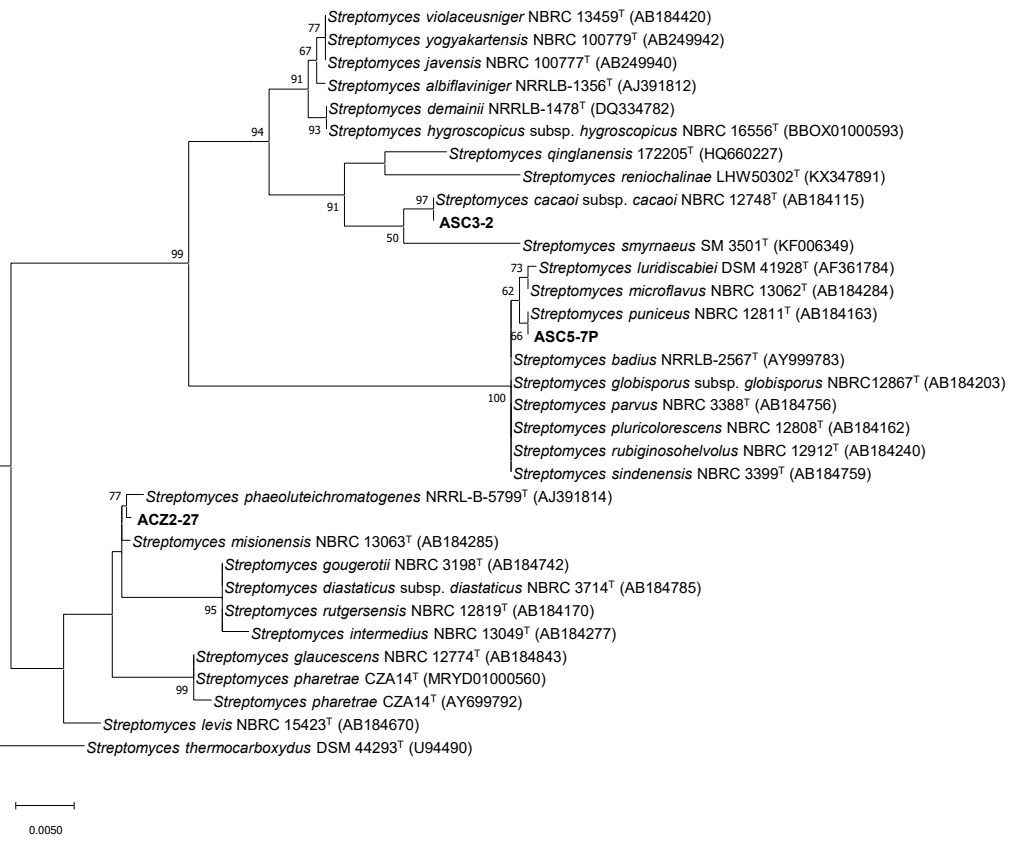

**Figure 2** **The Maximum likelihood tree of antagonistic *Streptomyces* strains ACZ2-27, ASC3-2 and ASC5-7P.** The Maximum likelihood tree based on 16S rRNA gene sequences (1,296 nucleotides) showing the phylogenetic position of antagonistic *Streptomyces* strains ACZ2-27, ASC3-2 and ASC5-7P relative to the type strains of other related *Streptomyces* species. *Streptomyces thermocarboxydus* DSM 44293[T] was used as an outgroup. The number at each node is the bootstrap support value (%) based on 1,000 replicates (only values >50% are shown). The bar shows 0.005 substitutions per nucleotide position.

closely related type strains was calculated using the pairwise alignment option in BioEdit program version 7.2.5. The results revealed that isolates ACZ2-27, ASC3-2 and ASC5-7P were closely related to *S. misionensis* NBRC 13063[T] (99.71%), *S. cacaoi* subsp. *cacaoi* NBRC 12748[T] (100%) and *S. puniceus* NBRC 12811[T] (100%), respectively. This is the first report of actinomycete species isolated from the black dwarf honey bee (*A. andreniformis*) that showed the ability to produce antimicrobial metabolites that inhibited the growth of the phytobacterial pathogen *X. campestris* pv. *campestris*.

In addition, the representative isolates of group I (AGA3-9, AGA3-53), group II (AGA3-58) and group III (ASC2-5) were identified by 16S rRNA gene sequence analysis. The 16S rRNA gene sequences of strains AGA3-9 (LC546088), AGA3-53 (LC546089), AGA3-58 (LC546090) and ASC2-5 (LC546091) were analyzed by BLAST using the EzBioCloud database. The representative isolates from group I-III exhibited high similarity with members of the genera *Micromonospora*, *Actinomadura* and *Nonomuraea*, respectively (Table 2). The 16S rRNA gene sequence similarity between each representative isolate

and their closely related type strains showed that group I (AGA3-9 and AGA3-53) were most closely related to *Micromonospora chalcea* DSM 43026$^T$ (99.34%) and *M. halotolerans* CR18$^T$ (99.70%), respectively. The representative isolates of group II (AGA3-58) and group III (ASC2-5) were closely related with *Actinomadura meyerii* DSM 44715$^T$ (99.34%) and *Nonomuraea salmonea* DSM 43678$^T$ (99.92%), respectively. The maximum likelihood tree also confirmed the placement of these actinomycetes to their respective genera (Fig. 3). This is the first evidence of rare actinomycetes associated with the black dwarf honey bee *A. andreniformis*.

## DISCUSSION

In this study, *Streptomyces* were found as the majority of the obtained isolates. In addition, members of rare actinomycete genera *Micromonospora*, *Nonomuraea* and *Actinomadura* were recovered from the black dwarf honey bee. All of these isolates were recovered from the adult honey bee except for isolate ASC5-7P which was obtained from the pollen. Actinomycetes belonging to the genera *Streptomyces*, *Nonomurea* and *Nocardiopsis* have been previously isolated from three species of honey bees (*A. mellifera*, *A. cereana* and *A. florea*) collected from northern Thailand (*Promnuan, Kudo & Chantawannakul, 2009*). Recently, *Streptomyces* spp. were also isolated from giant honey bee (*A. dorsata*) (*Promnuan, Promsai & Meelai, 2020*). These observations suggested that members of the genus *Streptomyces* may be commonly isolated from at least 5 species of honey bee in Thailand (*A. mellifera*, *A. cereana*, *A. florea*, *A. dorsata* and *A. andreniformis*). The evidence of rare actinomycetes associated with the black dwarf honey bee, *A. andreniformis*, implied that this honey bee species may harbor diverse actinomycete populations. It is also interesting to note that members of the genus *Micromonospora* were isolated from honey bee for the first time. Though, the function of these associated actinomycetes in *A. andreniformis* is still unknown, it is tempting to suggest that this relationship is at least neutral to the bees as all the bee samples used for isolation were healthy.

Actinomycetes are known for the production of bioactive compounds with antimicrobial activity against plant pathogens such as bacteria and fungi (*Viaene et al., 2016*). Several actinomycete species obtained from various habitats have been reported for secondary metabolites against phytopathogens. *Streptomyces* spp. obtained from marine samples in Egypt showed variable antimicrobial activity, secretion of numerous hydrolytic enzymes, *in vitro* and *in vivo* nematicidal activity against root-knot nematodes and supported plant growth (*Rashad et al., 2015*). *S. violaceusniger* strain A5 isolated from chitin-rich partially decomposed molted snake skin, showed strong inhibitory activity against *Xanthomonas axonopodis* pv. *punicae*, the causative agent of oily spot disease in pomegranate, with a MIC in the range of 0.625–1.25 mg mL$^{-1}$ (*Chavan et al., 2016*). Endophytic *Streptomyces* spp. (AB131-1 and AB131-2) reduced the infection of *X. oryzae* pv. *oryzae* a causal agent of bacterial leaf blight (BLB disease) and improved the growth of rice seedlings in a pot experiment in the greenhouse (*Hastuti et al., 2012*). *Streptomyces* isolated from compost-amended soil had antimicrobial activity against *Agrobacterium tumefaciens* (*Cuesta et al., 2010*). *S. caeruleatus* isolated from the *Cassia fistula* rhizosphere soil showed strong

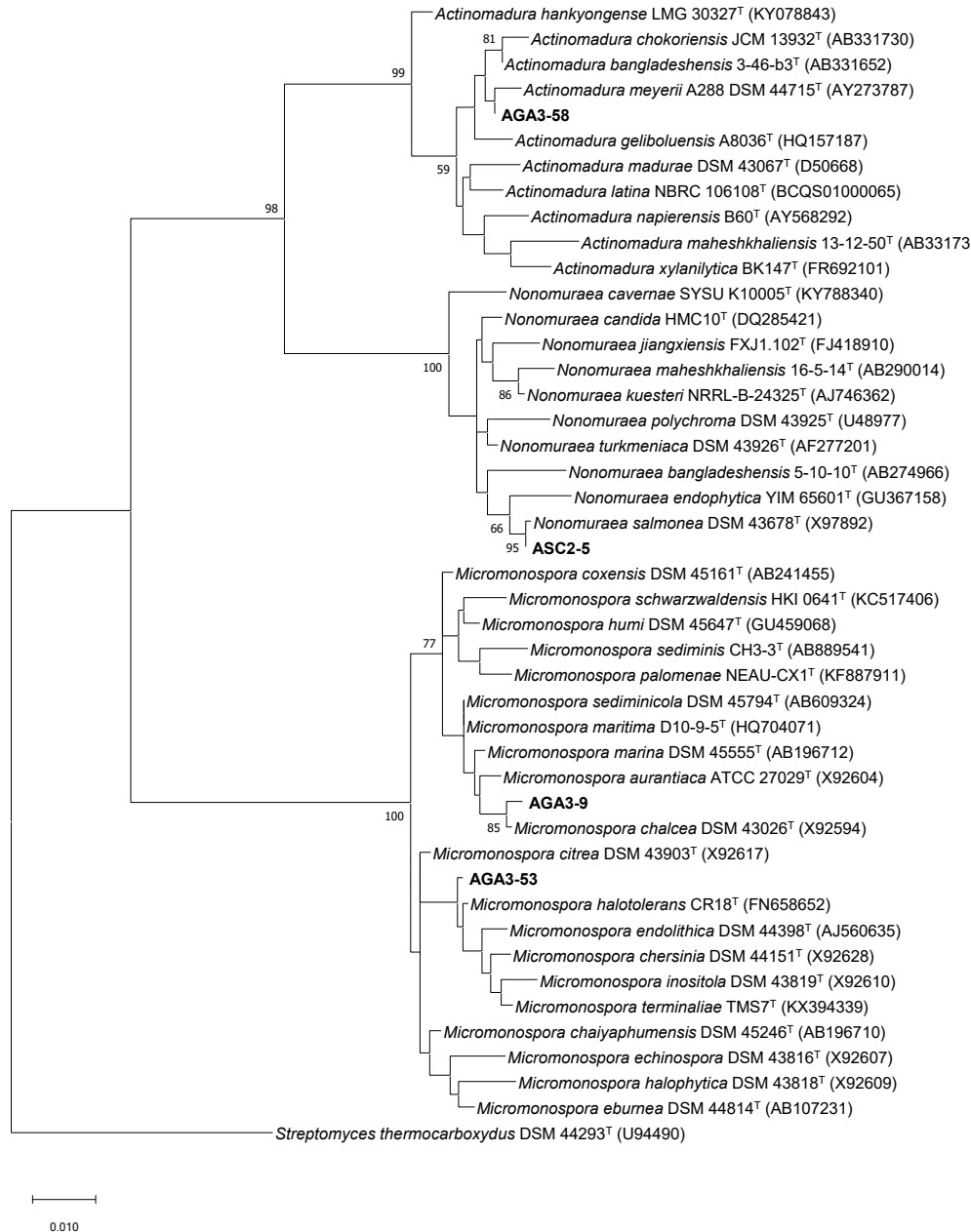

**Figure 3  The Maximum likelihood tree of the representative strains of non-*Streptomyces* isolates.** The Maximum likelihood tree based on 16S rRNA gene sequences (1,297 nucleotides) showing the phylogenetic position of the representative strains of non-*Streptomyces* isolates and their nearest neighbors. *Streptomyces thermocarboxydus* DSM 44293[T] was used as an outgroup. The number at each node is the bootstrap support value (%) based on 1,000 replicates (only values >50% are shown). The bar shows 0.01 substitutions per nucleotide position.

activity against *X. campestris* pv. *glycine*, the soybean pathogen (*Mingma et al., 2014*). It is evident from these studies and our results that *Streptomyces* are effective in control of plant pathogens including *Xanthomonas* species.

Several publications support the view that insects provide new sources of actinomycetes that may produce novel natural products with antimicrobial properties (*Chevrette et al., 2019*). *Streptomyces* spp. have been isolated from *A. andreniformis* and showed high activity in reducing the egg hatch rate, increasing the infective second-stage juvenile mortality rate of the root-knot nematode (*Meloidogyne incognita*) and reducing root gall of chili in a pot experiment (*Santisuk et al., 2018*). This study and our results indicated that the black dwarf honey bee (*A. andreniformis*) is an interesting new source for screening of actinomycetes that may produce novel natural products for agriculture. *Streptomyces* spp., *Nonomurea* spp. and *Nocardiopsis* spp. were isolated from honey bees (*A. mellifera*, *A. cereana* and *A. florea*) and showed antibacterial activity against the growth of *Paenibacillus larvae* and *Melisococcus plutonius* that cause American foulbrood and European foulbrood disease in honey bee (*Promnuan, Kudo & Chantawannakul, 2009*). The actinobacteria (*Pseudonocardia* spp.) associated with fungus-growing ants (*Apterostigma dentigerum*) produced dentigerumycin which inhibited the growth of parasitic fungus (*Escovopsis* sp.) (*Oh et al., 2009*). *Poulsen et al. (2011)* reported the novel macrocyclic lactam, sceliphrolactam compounds from *Streptomyces* symbionts with mud dauber wasps (*Chalybion californicum* and *Sceliphron caementarium*). *Streptomyces* spp., *Micromonospora* spp. and *Actinoplanes* spp. obtained from the paper wasp *Polistes dominulus* nests showed antibacterial activity against the growth of some pathogenic bacteria (*Pseudomonas aeruginosa*, *Escherichia coli*, *Staphylococcus aureus*, *Serratia marcescens* and *Bacillus subtilis*) (*Madden et al., 2013*). A novel compound cyphomycin was isolated from *Streptomyces* (ISID311) isolated from the microbiome of the fungus-growing ant which active against multidrug resistant fungal pathogens (*Chevrette et al., 2019*). *Matarrita-Carranza et al. (2021)* reported the genome sequence and the potential for antibiotic production of *Streptomyces* sp. M54, actinomycetes associated with the eusocial wasp (*Polybia plebeja*). This actinobacterium produces antimicrobial compounds that are active against *Hirsutella citriformis*, a natural fungal enemy of its host, and the human pathogens *Staphylococcus aureus* and *Candida albicans*. Recently, three actinomycetes (*S. ramulosus*, *S. axinellae* and *S. drozdowiczii*) isolated from giant honey bee (*A. dorsata*) combs showed the ability to inhibited the growth of black rot pathogen (*X. campestris* pv. *campestris*) with MIC value 32 mg L$^{-1}$ (*Promnuan, Promsai & Meelai, 2020*). These data indicated that actinomycetes associated with insects represent a valuable source for new antimicrobial compounds.

## CONCLUSIONS

Black rot disease, which is caused by *X.campestris* pv. *campestris*, can cause significant losses of the Brassicaceae in Asian countries. In this study, we provide evidence that actinomycetes including members of rare genera are associated with the black dwarf honey bee (*A. andreniformis*) especially in adult stage. The organic extract of isolate ACZ2-27 showed the lowest MIC of 64 mg L$^{-1}$ against the growth of black rot pathogen (*X. campestris* pv. *campestris*). This isolate was obtained from an adult bee and is phylogenetically closely related to *S. misionensis*. This is the first report of actinomycetes isolated from black dwarf honey bee which showed growth inhibitory activity against phytopathogenic bacterium

pathogen, *X. campestris* pv. *campestris.* Black dwarf honey bee associated actinomycetes may represent an interesting source for the search of bioactive compounds for biotechnological usage, especially in agriculture.

### Funding

This research was supported by the Department of Microbiology, Faculty of Liberal Arts and Science, Kasetsart University of the year 2018, the Research Promotion and Technology Transfer Center (RPTTC) of the Faculty of Liberal Arts and Science, Kasetsart University Kamphaeng Sean campus, Thailand, Grant SRIF-JRG-2562-04 from the Faculty of Science, Silpakorn University, Nakhon Pathom, Thailand and partially supported by Chiang Mai University. There was no additional external funding received for this study. The funders had no role in study design, data collection and analysis, decision to publish, or preparation of the manuscript.

### Grant Disclosures

The following grant information was disclosed by the authors:
Department of Microbiology, Faculty of Liberal Arts and Science, Kasetsart University (Kamphaeng Saen campus) of the year 2018.
The Research Promotion and Technology Transfer Center (RPTTC) of the Faculty of Liberal Arts and Science, Kasetsart University (Kamphaeng Sean campus).
Faculty of Science, Silpakorn University, Nakhon Pathom, Thailand: Grant SRIF-JRG-2562-04.
Chiang Mai University, Chiang Mai, Thailand.
Kasetsart University Kamphaeng Sean campus, Thailand.
The Faculty of Science, Silpakorn University, Nakhon Pathom: SRIF-JRG-2562-04.
Chiang Mai University.

### Competing Interests

The authors declare there are no competing interests.

### Author Contributions

- Yaowanoot Promnuan conceived and designed the experiments, performed the experiments, analyzed the data, prepared figures and/or tables, authored or reviewed drafts of the paper, and approved the final draft.
- Saran Promsai and Sujinan Meelai performed the experiments, authored or reviewed drafts of the paper, and approved the final draft.
- Wasu Pathom-aree analyzed the data, authored or reviewed drafts of the paper, and approved the final draft.

### Animal Ethics

The following information was supplied relating to ethical approvals (i.e., approving body and any reference numbers):

This study was approved by the Institutional Animal Care and the Use Committee (IACUC), Silpakorn University (Ethic number: 8603.16/0328).

## Field Study Permissions

The following information was supplied relating to field study approvals (i.e., approving body and any reference numbers):

Verbal permission was received from the farm owners (Mr. Ton Tatiya and Mr. Ma Madamun).

## DNA Deposition

The following information was supplied regarding the deposition of DNA sequences:

All sequences are available at GenBank: LC500236, LC506284, LC506285, LC546088, LC546089, LC546090 and LC546091.

## Data Availability

The Figs. S1 and S2 consist of multiple alignments of the closely related type strains of Streptomyces and non-Streptomyces.

## Supplemental Information

Supplemental information for this article can be found online at http://dx.doi.org/10.7717/peerj.12097#supplemental-information.

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
