# Peer review of "Apis andreniformis associated Actinomycetes show antimicrobial activity against black rot pathogen (Xanthomonas campestris pv. campestris)"

_PeerJ, doi:10.7717/peerj.12097_

## Round 0.1 · original submission · Major Revisions

Dear Dr. Promnuan and colleagues:

Thanks for submitting your manuscript to PeerJ. I have now received two independent reviews of your work, and as you will see, the reviewers raised some concerns about the research. Despite this, one reviewer is optimistic about your work and the potential impact it will have on research studying antimicrobial activity of Streptomyces spp. isolated from the black dwarf honey bee. Thus, I encourage you to revise your manuscript, accordingly, taking into account all of the concerns raised by the reviewer.

Importantly, please ensure that an English expert has edited your revised manuscript for content and clarity.

There are many specific issues pointed out by the reviewer, and you will need to address all of these and expect a thorough review of your revised manuscript by this same reviewer.

Good luck with your revision,

-joe

Reviewer 1 ·

Basic reporting

The authors have isolated and identified three Streptomyces strains associated with the black dwarf honey bee (Apis adreniformis) as well as several strains from underrepresented actinobacterial genera. The three Streptomyces strains were found to inhibit the growth of the Gram-negative plant pathogen Xanthomonas campestris pv. campestris in vitro. The authors also demonstrated that organic extracts prepared from the inhibitory Streptomyces strains were able to inhibit Xanthomonas campestris pv. campestris. The authors argue that the presence of inhibitory Streptomyces strains and rare actinobacterial genera isolated from this rare honey bee species may make it a valuable source for future natural product discovery efforts.

The English is mostly clear but should be improved in several sections throughout the paper prior to publication:
• Line 24 and 25 – to be more consistent, the word “with” should be removed and 84.4% should be put in parentheses.
• Line 31 – the word “a” should be added in “effect with a lower MIC…”
• Line 48 – The sentence “It is the 5th honey bee…” should simply say that it is “one of the known seven honey bee species”, unless there is some numbering scheme to rank them.
• Line 53 – The word “in” should be added in “distributed in nature both in terrestrial…”
• Line 55 – The word “bees” is repeated between leafcutter bees and stingless bees
• Line 58 – The word “belonged” should be changed to “belonging”
• Line 69 – The word “been” should be removed or should be changed to “has been declining”
• Line 70 – “search and discovery program” should be made plural “programs”
• Line 70 – The word “is” should be changed to “has”
• Line 78 – The word “problem” should be made plural; however, it would be more helpful to describe the actual problems to human health and the environment, such as antibiotic resistance and toxicity.
• Line 80 – “chemical products” should be changed to a more accurate word such as “antibiotics”
• Line 85 and 86 – I believe the sentence starting “Actinomycetes associated with…” is missing the words “with activity” or something similar before the word “against”
• Line 94 – The article “The” in “The permission was received…” should be removed
• Line 119 – The first “of” in the title should be changed to “for”, “Screening for antagonistic activity…”
• Line 132 – The word “store” should be in the past tense as “stored”
• Line 148 – “actinomycete” should be made plural
• Line 157 – The year in “Yoon et al.” should be “2017” not “20017”
• Line 171 – The word “actinomycete” should be plural
• Line 180 – The article “a” needs to be added in “growth of a phytobacterial”
• Line 196 – The word “program” can be removed
• Line 198 – The word “these” should be removed
• Line 198 – The word “their” should be changed to “the most”
• Line 213 – “were blasted” should be changed to “were analyzed by BLAST…”
• Line 217 – “shown” should be changed to “showed”
• Line 222 – The word “the” in “first evidence of the rare…” should be removed
• Line 228 – “belonged” should be changed to “belonging”
• Line 233 – The word “common” should be changed to “commonly isolated from” to reflect that these studies have all been isolation based
• Line 236 – “diverse actinomycetes population” should be “diverse actinomycete populations”
• Line 241 – The words “antibacterial” and “antiviral” should be changed to “bacterial” and “viral” as the authors are describing the pathogenic agents
• Line 244 – I just don’t know….
• Line 251 – The word “a” should be added to “in a pot experiment…”
• Line 257 – The sentence should be changed to say “are effective in the control of plant…”
• Line 259 – The words “for screening” should be removed or the sentence should be restructured.
• Line 286 – The sentence should say “obtained from an adult bee and is phylogenetically…”
• Line 289 – The word “This” should be removed
• Line 290 – There should be a comma between “usage” and “especially”

The much of the relevant research is cited; however, several of the papers that are cited are between twenty and thirty years old. In line 54, the findings are stated to be “recent”. There have been many more recent publications on the association of actinomycetes with fungus growing ants and leafcutter bees and they should be included. Additionally, in the discussion the authors state that there are several publications supporting their view that insects might be a source of new antimicrobials; however, they do not cite a recent publication documenting the antimicrobial potential of over 10,000 actinomycetes isolated from over 1,000 insect hosts that would greatly bolster their claim.
• The paper is a 2019 paper from Chevrette, M.G. et al. in Nature Communications.

The article conforms to the structure suggested by PeerJ and the figures and tables are relevant to the text in the manuscript.

All 16S sequences appear to be submitted to publicly available databases.

The results presented are relevant to the hypothesis being tested.

Experimental design

The research presented in the manuscript is within the aims and scope of PeerJ.

The research question is provided in the introduction and the relevance of the research is made clear. The authors state that this is the first study to isolate Streptomyces from the black dwarf honey bee in the abstract and the introduction; however, they later cite a paper from Santisuk et al, that reports isolation of Streptomyces spp. from the black dwarf honey bee with anti-nematode activity.
• The authors should make sure that they state that their study is the first to identify the antimicrobial potential of these Streptomyces, especially in the abstract.
• Additionally, the authors should mention in the introduction, perhaps at the end of the paragraph ending at line 71, why “rare genera” actinomycetes are of interest for drug discovery programs as well.

Replication was performed in each of the experiments and extra efforts were performed to confirm the MIC level by streaking the cultures from the microtiter plates onto agar plates. Recent versions of all bioinformatic tools were used in the production of the sequence alignments and phylogenetic trees.
• The MICs of the inhibitory Streptomyces organic extracts are all at the highest concentrations tested (128 and 64 mg/L). The MICs would be more accurate if the serial dilution was more evenly distributed around the MIC value.

The methods mostly rely on previously published protocols with minor adjustments that are described in the manuscript.
• A brief description of the extraction method should be provided in the manuscript, as the authors later, in line 136, describe the extract as an ethyl acetate extract without providing any context to how it was produced.
• It is unclear in line 100 if the five adult bees and five pupae were taken from each of the six collected combs or if it was five adult bees and five pupae total.
• Additionally, was an equivalent number of pollen and honey collections also sampled for actinomycetes?

Validity of the findings

The authors provide a table with information on the number of strains isolated from each source. Later in line 286, the authors describe isolate ACZ2-27 as being isolated from an adult bee.
• It would be very interesting to know where the other two inhibitory isolates (ASC3-2 and ASC5-7P) were isolated from, especially since the authors isolated from various sources in the honey bee environment.
• Additionally, it would be interesting to know where the “rare genera” actinobacteria were isolated from, or at least the representative strains. Did they come from unique isolation sources compared to the Streptomyces strains?
• The authors may also want to comment on any inhibitory activity observed in any of the “rare genera”.

Reviewer 2 ·

Basic reporting

English needs a lot of improvemet.

Experimental design

No Comment

Validity of the findings

No Comment.

Additional comments

No comment

---

## Round 0.2 · Minor Revisions

Dear Dr. Promnuan and colleagues:

Thanks for revising your manuscript. Reviewer 1 is mostly satisfied with your revision (as am I). Great! There are more edits to make and suggestions to consider. However, reviewer 2 has raised concern about the experiments needing to be repeated. Please address this ASAP (by rebuttal or further experiments).

Good luck with your revision,

-joe

Reviewer 1 ·

Basic reporting

The authors have isolated and identified three Streptomyces strains associated with the black dwarf honey bee (Apis adreniformis) as well as several strains from underrepresented actinobacterial genera. The three Streptomyces strains were found to inhibit the growth of the Gram-negative plant pathogen Xanthomonas campestris pv. campestris in vitro. The authors also demonstrated that organic extracts prepared from the inhibitory Streptomyces strains were able to inhibit Xanthomonas campestris pv. campestris. The authors argue that the presence of inhibitory Streptomyces strains and rare actinobacterial genera isolated from this rare honey bee species may make it a valuable source for future natural product discovery efforts.

The English is mostly clear but should be improved in several sections throughout the paper prior to publication:
• Line 49 – This sentence is repetitive and likely not necessary, since the other six species are never mentioned. The repetition should be removed or the sentence should be removed.
• Line 93 – The word “the” should be removed from “with the antagonistic…”
• Line 190 – The word “All” does not match with the percentages in line 186 and 187. Line 186 says that 65.6% of actinobacteria were isolated from adult bees, not all.
• Line 240 – the word “was” should be “were”
• Line 241 – the word should be “Actinomycete”, not plural
• Line 248 – There should be a space between the words “from” and “at”
• Line 250 – There should be a comma after “adreniformis”
• Line 255 – The word “are” should be changed to “were”
• Line 257 – The words should be “bacteria and fungi” not “bacterial and fungal agents” as you are describing what the compounds with activity are acting on.
• Line 260 – This sentence is difficult to read, it might be easier to read if you change it to “variable antimicrobial activity” instead of “antimicrobial activity with a variable spectrum”
• Line 268 and Line 270 – If all isolates in the study were determined to be Streptomyces, as it says in line 270, then “Actinomycetes” can simply be changed to “Streptomyces” and the sentence starting on 269 can be eliminated in order to simplify the sentence.
• Line 300 – A space should be between “albicans.” and “Recently”
• Line 310 – The “organic extract” of isolate ACZ2-27 had the lower MIC, not the isolate itself.

The much of the relevant research is cited and is relevant to the study.
The article conforms to the structure suggested by PeerJ and the figures and tables are relevant to the text in the manuscript.
All 16S sequences appear to be submitted to publicly available databases.
The results presented are relevant to the hypothesis being tested.

Experimental design

The research presented in the manuscript is within the aims and scope of PeerJ.

The research question is provided in the introduction and the relevance of the research is made clear. The authors state that this is the first study to isolate Streptomyces and other rare actinobacterial genera from the black dwarf honey bee and suggest these may be useful in future natural product studies.

Replication was performed in each of the experiments and extra efforts were performed to confirm the MIC level by streaking the cultures from the microtiter plates onto agar plates. Recent versions of all bioinformatic tools were used in the production of the sequence alignments and phylogenetic trees.

Validity of the findings

The conclusions are well stated, and the authors provide evidence for the novelty of their findings. The final sentence in the abstract; however, is not true.

• The authors state at the end of the abstract that “This is the first report to identify the antimicrobial potential of Streptomyces especially the antibacterial activity against phytopathogenic bacteria.” This claim is shown to be untrue in lines 256-274 that detail many studies on the antimicrobial activity of Streptomyces against phytopathogens in the same genus as the one being studied here. The authors either need to qualify the statement or remove it from the manuscript.

Reviewer 2 ·

Basic reporting

The article seems to be well written. However has major technical errors.

Experimental design

There are technical limitations.

Validity of the findings

-

Additional comments

The antibacterial assays are not understandable. The experiments need to be repeated.

---

## Round 0.3 · accepted · Accept

Dear Dr. Promnuan and colleagues:

Thanks for revising your manuscript based on the concerns raised by the reviewer. I now believe that your manuscript is suitable for publication. Congratulations! I look forward to seeing this work in print, and I anticipate it being an important resource for groups studying antimicrobial activity of Streptomyces spp. isolated from the black dwarf honey bee. Thanks again for choosing PeerJ to publish such important work.

Best,

-joe

Reviewer 1 ·

Basic reporting

See below

Experimental design

See below

Validity of the findings

See below

Additional comments

I believe the changes to the manuscript are acceptable and that the manuscript, once edited for English language, should be accepted for publication.